# Polypyrimidine tract binding protein 1 protects mRNAs from recognition by the nonsense-mediated mRNA decay pathway

**Zhiyun Ge[1], Bao Lin Quek[2], Karen L Beemon[2], J Robert Hogg[1]\***

[1]Biochemistry and Biophysics Center, National Heart, Lung, and Blood Institute, National Institutes of Health, Bethesda, United States; [2]Department of Biology, Johns Hopkins University, Baltimore, United States

**Abstract** The nonsense-mediated mRNA decay (NMD) pathway degrades mRNAs containing long 3'UTRs to perform dual roles in mRNA quality control and gene expression regulation. However, expansion of vertebrate 3'UTR functions has required a physical expansion of 3'UTR lengths, complicating the process of detecting nonsense mutations. We show that the polypyrimidine tract binding protein 1 (PTBP1) shields specific retroviral and cellular transcripts from NMD. When bound near a stop codon, PTBP1 blocks the NMD protein UPF1 from binding 3'UTRs. PTBP1 can thus mark specific stop codons as genuine, preserving both the ability of NMD to accurately detect aberrant mRNAs and the capacity of long 3'UTRs to regulate gene expression. Illustrating the wide scope of this mechanism, we use RNA-seq and transcriptome-wide analysis of PTBP1 binding sites to show that many human mRNAs are protected by PTBP1 and that PTBP1 enrichment near stop codons correlates with 3'UTR length and resistance to NMD.

**\*For correspondence:** j.hogg@nih.gov

**Competing interests:** The authors declare that no competing interests exist.

## Introduction

Nonsense-mediated mRNA decay (NMD) is an evolutionarily conserved co-translational mRNA turnover pathway responsible for degrading diverse eukaryotic mRNAs (reviewed in *Schweingruber et al., 2013*). In addition to its well-known role in detecting aberrant transcripts containing premature termination codons (PTCs), NMD has also been shown to target a broad range of cellular mRNAs under normal conditions (*He et al., 2003*; *Hurt et al., 2013*; *Lelivelt and Culbertson, 1999*; *Mendell et al., 2004*; *Rehwinkel et al., 2005*; *Weischenfeldt et al., 2008*; *Weischenfeldt et al., 2012*; *Longman et al., 2013*), including those with spliced introns downstream of the TC, upstream open reading frames (uORFs), or long 3' untranslated regions (3'UTRs). Together, the pathway is commonly estimated to regulate 5–10% of all human genes. NMD thus has an enormous impact on the human transcriptome in all cells, while also performing a surveillance role that modulates the phenotypic consequences of numerous human genetic diseases arising from nonsense mutations (*Keeling et al., 2013*).

A core set of dedicated NMD proteins works in conjunction with general mRNA metabolism and translation factors to accurately select and degrade specific mRNAs. The superfamily I RNA helicase UPF1 is the focal point of the NMD pathway, with characterized functions throughout the process of target discrimination, translational repression, RNase recruitment, and post-decay RNP remodeling (*Czaplinski et al., 1995*; *Franks et al., 2010*; *Hogg and Goff, 2010*; *Isken et al., 2008*; *Okada-Katsuhata et al., 2012*; *Unterholzner and Izaurralde, 2004*). UPF2 and UPF3B stimulate UPF1's ATPase and decay-promoting activities and serve as a physical link between UPF1 and the exon junction complex (EJC), a strong activator of decay in vertebrate NMD (*Chamieh et al., 2008*; *Le Hir et al., 2001*). Interaction of UPF1 with release factors eRF1/3 at the terminating ribosome promotes

**eLife digest** Genes are used as templates to create molecules of messenger RNA (mRNA) that contain all the information needed to make a protein. This information begins with a 'start site' and ends with a 'stop site.' The regions of the mRNA outside of the start and stop sites are called untranslated regions.

Not all mRNAs are correctly made, and cells combat this problem by detecting and destroying faulty mRNAs before they are translated into protein. One way cells do this is by recognizing and destroying mRNAs that include long untranslated regions, which can indicate that the mRNA might have a stop site too early in its sequence. A key problem with this mechanism, however, is that long untranslated regions also serve important roles in the cell: for example, by determining where and when mRNA molecules are read to make protein. How then do mRNAs with long but important untranslated regions escape detection and degradation?

Ge et al. have now investigated this question using an approach that allows a 'handle' to be attached to particular RNA molecules. This allows the RNA and any proteins bound to it to be purified away from all other RNAs and proteins in the cell, and the proteins can then be identified by a technique called mass spectrometry.

Ge at al. found that mRNAs can recruit a protein called PTBP1 to part of the RNA sequence near the stop site. This prevents an RNA decay protein recognizing and triggering the degradation of the mRNA, even if the mRNA has a long untranslated region.

Thus, PTBP1 plays a crucial role in protecting human RNAs with long untranslated regions from destruction by the nonsense-mediated decay pathway. Some viral RNAs are also able to evade decay, and so Ge et al. hypothesize that the virus stole this method for maintaining its RNAs from host cells. A future goal is to understand whether this system works the same way in all cell types or protects different RNAs in different cells.

phosphorylation of UPF1 by the SMG1 kinase, which in turn mediates recruitment and/or activity of the SMG6 endonuclease, the SMG5/7 complex, and decapping and deadenylation factors (*Jonas et al., 2013*; *Loh et al., 2013*; *Chakrabarti et al., 2014*; *Glavan et al., 2006*; *Huntzinger et al., 2008*; *Kashima et al., 2006*; *Nicholson et al., 2014*; *Ohnishi et al., 2003*; *Okada-Katsuhata et al., 2012*; *Eberle et al., 2009*; *Unterholzner and Izaurralde, 2004*).

The UPF1 helicase domain forms an extensive surface for high-affinity, sequence-nonspecific RNA binding, allowing the protein to associate with the full diversity of NMD substrates (*Chakrabarti et al., 2011*; *Fiorini et al., 2012*; *Hurt et al., 2013*; *Zund et al., 2013*; *Kurosaki et al., 2014*). Elongating ribosomes can disrupt UPF1's promiscuous RNA binding activity, leading to preferential UPF1 accumulation on 3'UTRs (*Hogg and Goff, 2010*; *Hurt et al., 2013*; *Zund et al., 2013*). We have previously proposed that this property of UPF1 effectively allows it to sense 3'UTR length, predisposing transcripts containing long 3'UTRs to decay (*Hogg, 2011*; *Hogg and Goff, 2010*). This model is supported by recent comprehensive analysis of UPF1 targets demonstrating correlations among mRNA 3'UTR length, UPF1 association, and de-repression of gene expression following NMD inhibition (*Hurt et al., 2013*).

Extensive experiments in several eukaryotic model systems have established a tight link between 3'UTR length and decay susceptibility, suggesting a conserved mechanistic basis for NMD target selection (*Amrani et al., 2004*; *Behm-Ansmant et al., 2007*; *Buhler et al., 2006*; *Eberle et al., 2008*; *Longman et al., 2007*; *Muhlrad and Parker, 1999*; *Singh et al., 2008*). Directly probing the relationship between 3'UTR length and NMD-sensitivity in human cells, systematic analyses of reporter transcripts revealed a progressive UPF1-dependent decrease in mRNA half-life with increasing 3'UTR length (*Buhler et al., 2006*; *Eberle et al., 2008*). While global correlations between 3'UTR length and NMD susceptibility have been described in mammals (*Hansen et al., 2009*; *Hurt et al., 2013*; *Mendell et al., 2004*; *Ramani et al., 2009*; *Yepiskoposyan et al., 2011*), these studies have been complicated by the fact that a large number of eukaryotic transcripts with long 3'UTRs appear to evade NMD through as-yet-unknown mechanisms. This is particularly true in human cells, in which the average 3'UTR length has evolved to exceed that necessary to induce

NMD of reporter transcripts. Indeed, several TC-proximal human mRNA sequences capable of protecting mRNAs from NMD have been identified (*Toma et al., 2015*). Thus, mechanisms to protect specific mRNAs from NMD are likely to be a major force shaping human gene expression.

Retroviruses have long served as excellent model systems to explore the functions of host mRNA processing, translation, and decay pathways. As a consequence of evolutionary pressures to maximize the coding potential of compact RNA genomes, retroviral RNAs possess features that are predicted to be targeted by host cell RNA surveillance machineries, including long 3'UTRs, multiple open reading frames, and retained introns (*Bolinger and Boris-Lawrie, 2009*; *Withers and Beemon, 2011*). Consequently, some retroviruses have developed protein-based protective mechanisms against NMD to ensure RNA integrity and translation (*Mocquet et al., 2012*; *Nakano et al., 2013*). Of known cellular or viral mechanisms for NMD evasion, the most extensively characterized is a cis-acting RNA sequence found in the Rous sarcoma virus (RSV), designated the RNA stability element (RSE; *Arrigo and Beemon, 1988*; *Barker and Beemon, 1991*, *1994*). The RSE is a 400-nt element located immediately downstream of the *gag* TC in the unspliced RSV viral RNA that robustly protects the viral RNA from UPF1-dependent decay in chicken cells (*Quek and Beemon, 2014*; *Weil and Beemon, 2006*; *Weil et al., 2009*; *Withers and Beemon, 2010*, *2011*). Despite thorough studies of the RSE structure and function, its mechanism of action has remained unclear.

Here, we elucidate the mechanism underlying the ability of the RSE to protect mRNAs from NMD and show that numerous human transcripts containing long 3'UTRs also exploit this strategy to maintain stability. By affinity purifying endogenously assembled mRNPs containing the RSE, we identify the polypyrimidine tract binding protein 1 (PTBP1) as the key mediator of RSE function. We show that mutations preventing PTBP1 binding to the RSE abolish protection from NMD, while artificial recruitment of PTBP1 immediately downstream of an NMD-triggering TC recapitulates RSE activity. Together, our findings indicate that PTBP1 functions to exclude UPF1 from 3'UTRs, disrupting its ability to accurately discriminate 3'UTR length and induce decay. Furthermore, we performed RNA-seq analysis on human cells depleted of PTBP1 and UPF1 together and in isolation to identify endogenous transcripts with long 3'UTRs protected from NMD by PTBP1. Transcriptome-wide analysis of PTBP1 interaction sites reveals preferential binding of PTBP1 near TCs, a binding pattern correlated with 3'UTR length and resistance to NMD.

## Results

### The RSE protects mRNAs from NMD in human cells

We first set out to test whether the avian retrovirus-derived RSE retains its anti-NMD function in human cells, reasoning that the ability to function in highly divergent vertebrates would imply a conserved mechanism for mRNA stabilization. For these studies, we used tetracycline (tet)-regulated reporter mRNAs containing a β-globin mini-gene and the SMG5 3'UTR (*Singh et al., 2008*). The SMG5 3'UTR triggers NMD as part of an extensive program of autoregulation by the NMD pathway, in a manner proposed to be due to its length (1342 nt; *Huang et al., 2011*; *Singh et al., 2008*; *Yepiskoposyan et al., 2011*). We inserted the 400 nt RSE sequence or a control sequence of the same length (the antisense RSE sequence, AS-RSE) into the reporter mRNAs immediately downstream of the TC, mimicking the natural context of the RSE in the RSV RNA (*Figure 1A,B*). To assess the specific antagonistic activity of the RSE against NMD, constructs encoding reporter mRNAs were co-transfected with a vector constitutively expressing a control RNA into 293 Tet-off cells treated with control or anti-UPF1 siRNAs. The expression of the tet-regulated mRNAs was induced for 4 hr before transcription was inhibited by addition of doxycycline, and mRNA decay was monitored at the indicated time points (*Figure 1C*). Transcripts containing only the SMG5 3'UTR exhibited a half-life of ~120 min in cells treated with control siRNAs, in agreement with its previous characterization as an NMD substrate. Transcripts containing the RSE were substantially more stable (half-life ~400 min) than transcripts containing the AS-RSE sequence (<120 min), confirming the protective activity of the RSE (*Figure 1C*, upper panel). In contrast, in cells depleted of UPF1, all transcripts had half lives of greater than 240 min, indicating that the observed decay in siNT-treated cells was due to the activity of UPF1 (*Figure 1C*, lower panel).

Previous studies of the RSE have shown that its activity is strongly position-dependent. Specifically, the RSV viral RNA becomes unstable when a TC is created upstream of the natural *gag* stop

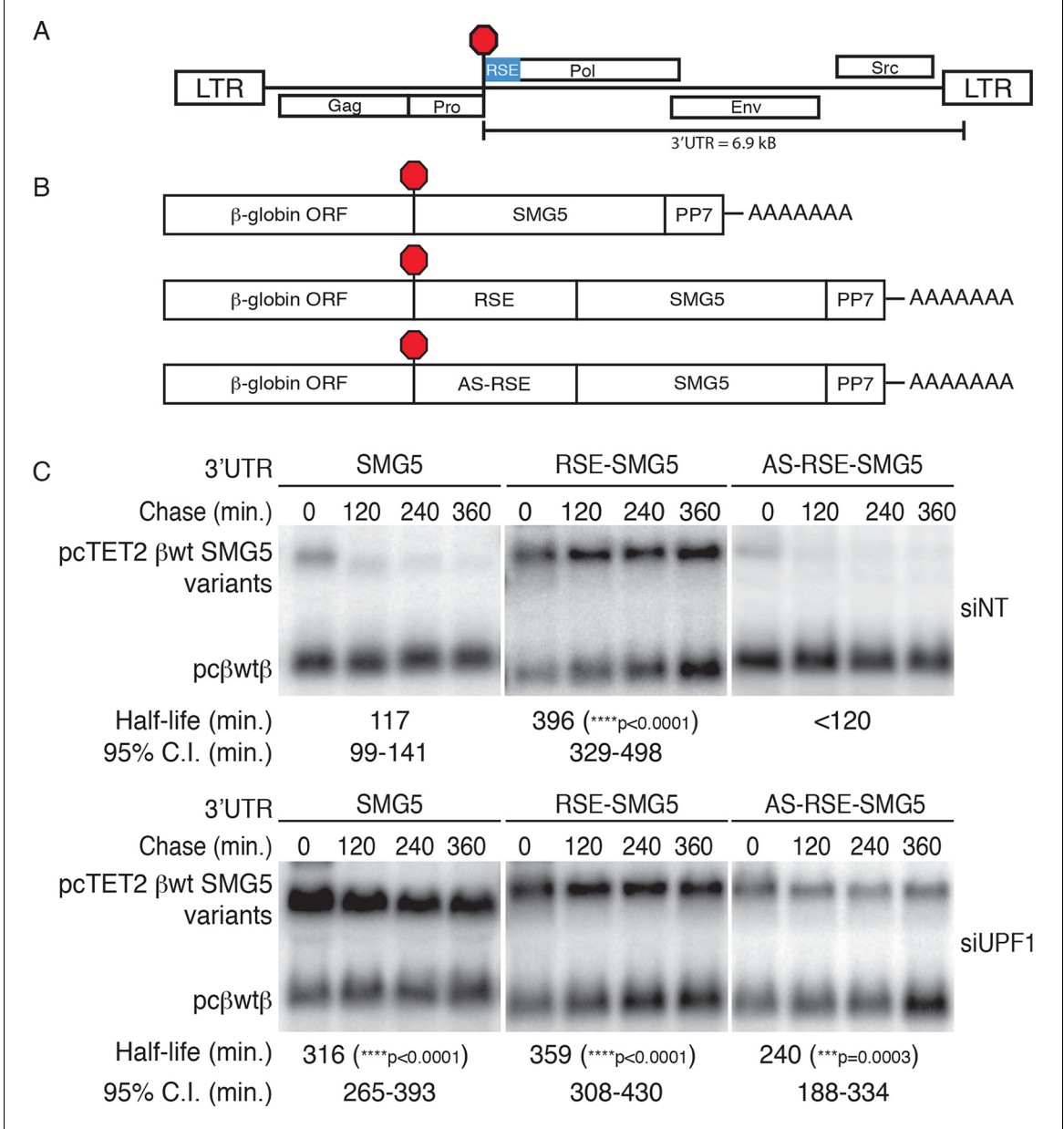

**Figure 1.** The RSE protects reporter mRNA from NMD in mammalian cells. (**A**) Schematic of the Rous sarcoma proviral genome. The RSE is located immediately downstream of the gag stop codon. (**B**) Schematic of tet-regulated β-globin reporter mRNA constructs used in RNA decay assays. The RSE sequence (middle) and a control sequence, the antisense RSE (AS-RSE) sequence (bottom), were inserted into reporter mRNAs containing the β-globin gene and the human SMG5 3'UTR (top). (**C**) Decay assays of reporter mRNAs containing the wild-type SMG5 3'UTR or variants supplemented with RSE or AS-RSE sequences. 293 Tet-off cells were treated with non-targeting siRNA (siNT; upper panel) or UPF1 siRNA (siUPF1; lower panel). Constructs encoding the indicated tet-regulated transcripts were co-transfected with the constitutively expressed wild-type β-globin reporter (pcβwtβ; bottom bands). Remaining RNA levels at indicated time points were normalized to levels of the wild-type β-globin transfection control. Half-lives and 95% confidence intervals were obtained from 3 independent experiments (***p<0.001; ****p<0.0001 in two-tailed ANCOVA analysis when compared to pcTET2-βwt-SMG5). Rapid decay of AS-RSE mRNAs to background levels in siNT samples precluded accurate quantification of decay rate. See also *Figure 1—figure supplements 1* and *2*.

The following figure supplements are available for figure 1:

**Figure supplement 1.** RSE protective activity is position-dependent.

**Figure supplement 2.** The RSE protects reporter mRNAs from EJC-stimulated NMD

codon, an effect due to the increased distance between the RSE and the PTC (*Barker and Beemon, 1994*; *Weil and Beemon, 2006*; *Withers and Beemon, 2011*). Here, we confirmed that the RSE also exhibits this property when used to shield reporter transcripts in human cells, as placement of the RSE in the middle or at the end of the SMG5 3′UTR abolished its protective activity (*Figure 1—figure supplement 1*). Together, our data suggest that the mechanisms governing RSE-mediated protection from NMD are conserved from birds to humans.

While the physiological mRNAs subject to RSE-mediated protection are unspliced retroviral RNAs, we also tested whether the RSE is capable of inhibiting NMD promoted by the presence of a 3′UTR-bound EJC. We inserted the wild-type RSE or antisense RSE sequences upstream of a previously characterized artificial GAPDH-derived 3′UTR engineered to contain the adenovirus major-late intron (AdML; *Singh et al., 2008*; *Figure 1—figure supplement 2*). As expected, the intron-containing 3′UTR caused rapid decay of tet-regulated reporter constructs. In contrast, mRNAs containing the RSE were highly stabilized, indicating that the RSE is capable of suppressing both EJC-independent and EJC-stimulated NMD.

## The RSE limits UPF1 association with mRNAs

Since UPF1 recruitment is a prerequisite for NMD, we hypothesized that the RSE may employ the simple strategy of preventing UPF1 from associating with the 3′UTR of the protected transcript. To test this idea, we immunopurified endogenous UPF1 and assayed the recovery of reporter mRNAs containing RSE or control sequences placed upstream of the artificial 3′UTR derived from the GAPDH ORF and 3′UTR, in this case lacking additional intronic sequence (*Figure 2A*). This GAPDH-derived 3′UTR has been previously used for studies of UPF1 association and decay and can be efficiently protected by a TC-proximal RSE (*Figure 2—figure supplement 1*; *Hogg and Goff, 2010*; *Singh et al., 2008*). Because UPF1 associates with mRNAs in a 3′UTR length-dependent manner (*Hogg and Goff, 2010*; *Kurosaki and Maquat, 2013*), we used an NMD-permissive 397 nt fragment of the SMG5 3′UTR (SMG5-397) to equalize 3′UTR lengths among the mRNAs studied and thus allow accurate assessment of the influence of the RSE on UPF1 binding. As additional controls, we tested the ability of UPF1 to bind mRNAs in which the RSE or the SMG5-397 fragments were moved to the 3′ end of the mRNA (3′-RSE and 3′-SMG5-397, respectively).

Plasmids encoding the experimental RNAs were co-transfected with a control plasmid producing mRNA containing only the GFP ORF and the bovine growth hormone polyadenylation element (bGH), which associates with UPF1 at low but consistent levels due to its short 3′UTR and serves as an internal standard for RNA recovery throughout immunoprecipitation and RNA isolation (*Hogg and Goff, 2010*). The extent of UPF1 binding was then assayed by northern blotting of the reporter mRNAs co-immunopurified with endogenous UPF1. As shown in *Figure 2B and C*, we observed a marked reduction in UPF1 association with RSE-containing RNAs relative to RNAs in which the RSE was moved to the 3′ end of the transcript or replaced with the SMG5-397 element. Together, these data indicate that the RSE prevents NMD by reducing UPF1 binding to 3′UTRs in a sequence- and position-dependent manner (*Figure 2B,C*).

While previous studies have shown that UPF1 association with mRNAs precedes commitment to decay and does not require ongoing translation (*Hogg and Goff, 2010*; *Hurt et al., 2013*; *Zund et al., 2013*), it remained possible that a reduction in UPF1 binding to RSE-containing 3′UTRs was a consequence rather than a cause of NMD inhibition. To rule out this possibility, we performed immunoprecipitations using extracts from cells treated with the chain-terminating translation inhibitor puromycin. Indeed, the RSE retained the capacity to antagonize UPF1 binding in the absence of translation termination events (*Figure 2—figure supplement 2*). These data suggest that the position of the RSE in the mRNA with respect to the 5′ and 3′ ends may contribute to its ability to disrupt UPF1 binding to mRNAs.

## Identification of proteins associated with RSE-containing mRNAs

To identify possible protein cofactors of the RSE, we isolated mRNPs containing the RSE in the sense or antisense orientations by an RNA-based affinity purification technique (*Hogg and Goff, 2010*). Reporter mRNAs tagged with a single copy of the 25 nt *Pseudomonas* phage 7 coat protein (PP7CP) RNA hairpin binding site (*Figure 3—figure supplement 1A*) were expressed in 293T cells, and the resulting endogenously assembled mRNPs were purified from cell extracts using the PP7CP

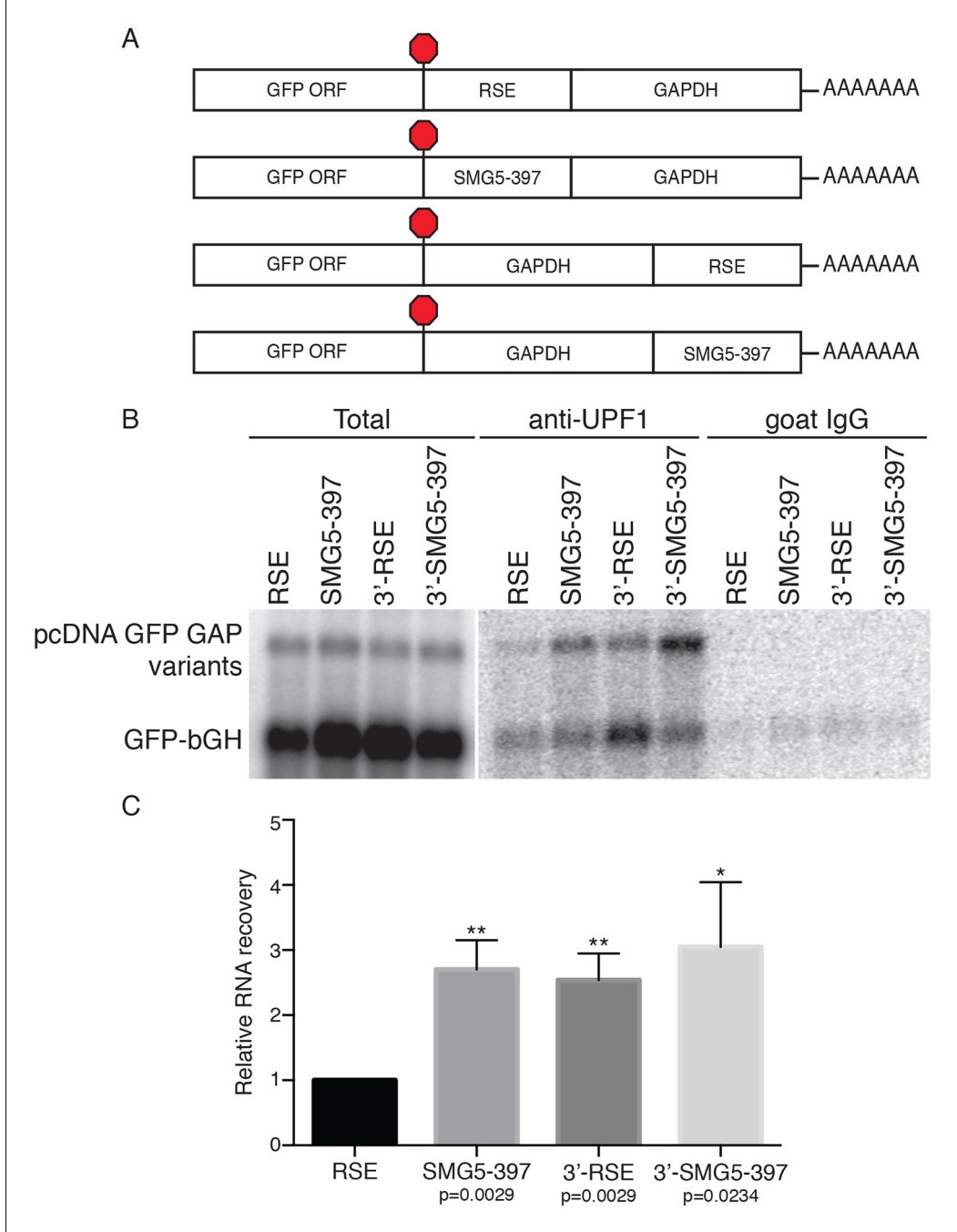

**Figure 2.** The RSE reduces UPF1 association with 3'UTRs in a position-dependent manner. (**A**) Schematic of reporter mRNA constructs used in immunoprecipitation assays. RNAs containing the GFP ORF and an artificial NMD-inducing 3'UTR comprising a portion of the human GAPDH ORF and the GAPDH 3'UTR (*Singh et al., 2008*) were modified to contain the RSE or the SMG5-397 sequence 5' or 3' of the GAPDH sequence. (**B**) Upf1 is reduced on 3'UTRs containing a TC-proximal RSE. Plasmids expressing the indicated mRNAs were co-transfected in 293 cells with a construct expressing the GFP ORF followed by the bovine growth hormone (bGH) polyadenylation signal. Endogenous UPF1 was immunoprecipitated from transfected cells, and co-purifying mRNAs were analyzed by northern blot. Bulk goat IgG was used as a non-specific interaction control. (**C**) Quantification of relative RNA recovery upon UPF1 immunoprecipitation, normalized to the recovery of co-transfected GFP-bGH mRNAs. The amount of RSE-containing RNA recovered was set to 1. Error bars indicate ± SD; n = 3 (*p<0.05; **p<0.01 in two-tailed Student's t-tests when compared to RSE recovery). See also *Figure 2—figure supplements 1* and *2*.

*Figure 2 continued on next page*

*Figure 2 continued*

The following figure supplements are available for figure 2:

**Figure supplement 1.** The RSE protects reporter mRNAs containing a GAPDH-derived NMD-sensitive 3'UTR.

**Figure supplement 2.** RSE inhibition of UPF1 binding is independent of translation.

tagged with tandem *Staphylococcus aureus* protein A domains. The isolated mRNPs containing either the RSE sequence or the control sequence displayed a similar profile of co-purifying proteins, while mock purifications from cell extracts containing no tagged RNA yielded very few contaminating proteins (*Figure 3—figure supplement 1B*, additional data not shown). Comprehensive analysis of the composition of the purified mRNP complexes by tandem mass spectrometry revealed a large number of proteins that were equally represented in the RSE-containing mRNPs and the control mRNPs, many of which are common RNA binding proteins, ribosomal proteins, and translation factors (*Figure 3—source data 1*). UPF1 was enriched in the control mRNP (*Figure 3—figure supplement 1C*), consistent with our finding that the RSE inhibits UPF1 association with the 3'UTR (*Figure 2*). Several proteins were over-represented in mRNPs containing the RSE sequence, including polypyrimidine tract binding protein 1 (PTBP1), heterogeneous nuclear ribonucleoprotein L (hnRNP L), MATRIN 3, and splicing factor, arginine and serine-rich 14 (SFRS14; *Figure 3—figure supplement 1C*). We were able to confirm the preferential interaction between these proteins and the RSE by immunoblotting of purified mRNPs or co-immunoprecipitation experiments (*Figure 3* and additional data not shown).

## A prominent role for PTBP1 in RSE-mediated NMD inhibition

Among the putative RSE-binding proteins identified, we considered PTBP1 to be a promising putative anti-NMD factor for several reasons. In addition to its established role in alternative splicing, PTBP1 has been implicated as an RNA stability factor (reviewed in *Romanelli et al., 2013*). Moreover, as expected for a protein responsible for protecting specific mRNAs, PTBP1 has been found to be preferentially associated with long 3'UTRs (*Gama-Carvalho et al., 2006*). Finally, a variety of techniques have been used to characterize the interaction specificity of PTBP1 and establish that the protein's four RNA recognition motifs (RRMs) each contribute to binding of degenerate pyrimidine-rich sequences. UV cross-linking and immunoprecipitation (CLIP) experiments indicate that PTBP1 binding is driven both by high-affinity heptameric or hexameric motifs and by local enrichment of pyrimidines (*Xue et al., 2009*; *2013*; *Han et al., 2014*). Indicative of the potential for robust PTBP1 binding, examination of the RSE sequence revealed as many as eleven CU-rich clusters corresponding to putative PTBP1 binding sites spread throughout the 400 nt element. This includes three instances of CU-rich hexameric sequences identified as over-represented in PTB CLIP sequence tags, as well as five additional clusters containing heptamers enriched in CLIP tags (*Xue et al., 2009*; see Materials and methods for sequences). Therefore, we predicted that multiple PTBP1 molecules might be recruited to RSE-containing mRNPs, forming an RNP refractory to UPF1 association.

To test this idea, we constructed two RSE sequence variants. First, we mutated all putative PTBP1 binding sites in the RSE by replacing a subset of U and C residues in CU-rich clusters with G and A residues, respectively (RSE-ΔPTB; see Materials and methods for sequences). Next, we inserted six previously characterized PTBP1 binding sites into the RSE-ΔPTB sequence at arbitrary positions (*Figure 3A*; *Xue et al., 2009*), with ~60 nt between each site (RSE-ΔPTB+6xPTBBS). The wild-type RSE and the RSE mutants were inserted into reporter mRNAs upstream of the artificial GAPDH-derived 3'UTR, and the extent of PTBP1 association with the reporter transcripts was assayed by monitoring the co-purification of reporter mRNAs with immunopurified endogenous PTBP1. As shown in *Figures 3B and C*, PTBP1 association with the RSE-ΔPTB-containing transcripts was significantly reduced compared to the wild-type RSE-containing transcripts. Reintroducing six PTBP1 binding sites into the RSE-ΔPTB led to recovery of PTBP1 association to levels similar to those found in association with the wild-type RSE.

We next investigated whether UPF1 binding was affected by eliminating the putative PTBP1 binding sites in the RSE sequence. To test the possibility that PTBP1 binding causes exclusion of UPF1 from the mRNP, we assayed the co-purification of the same reporter mRNAs with immunopurified

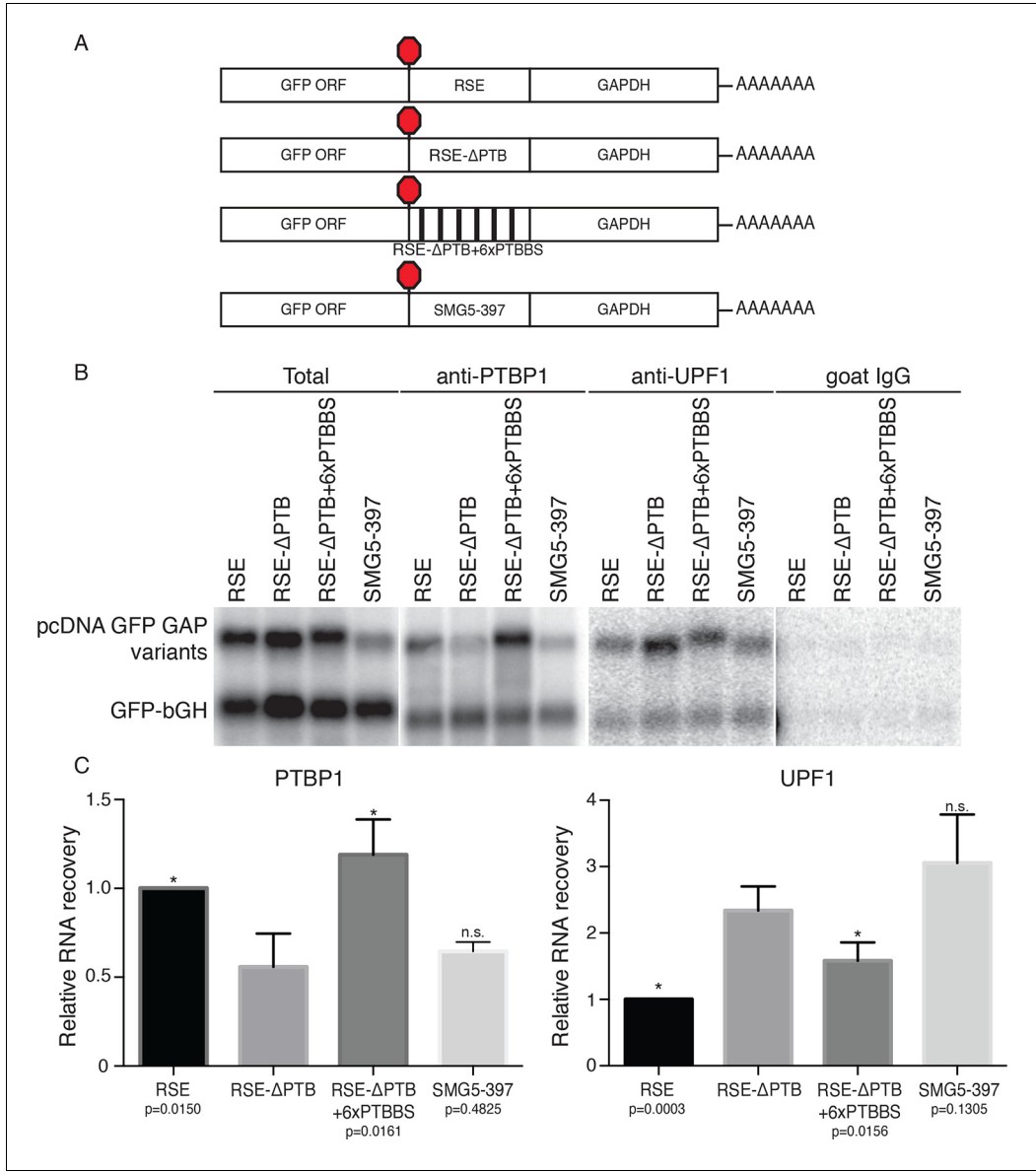

**Figure 3.** Accumulation of PTBP1 on the 3'UTR prevents UPF1 binding. (**A**) Schematic of reporter mRNA constructs used in immunoprecipitation assays. (**B**) PTBP1 is reduced on transcripts containing the RSE mutants lacking the putative PTBP1 binding sites. Immunoprecipitations were performed as in *Figure 2B*. Goat IgG samples were imaged using identical settings to those used for anti-UPF1 samples. (**C**) Left Panel: Quantification of relative RNA recovery upon PTBP1 immunoprecipitation, normalized to the recovery of co-transfected GFP-bGH mRNAs. Right Panel: Quantification of relative RNA recovery upon UPF1 immunoprecptation, normalized to the recovery of co-transfected GFP-bGH mRNAs. The amount of RSE-containing RNA recovered was set to 1. Error bars indicate ± SD; n ≥ 3 (*p<0.05 in two-tailed Student's t-tests when compared to RSE-ΔPTB recovery. See also *Figure 3—figure supplement 1* and *Figure 3—source data 1*.

The following source data and figure supplement are available for figure 3:

**Source data 1.** Table of raw mass spectrometry data.

**Figure supplement 1.** Identification of RSE-interacting proteins by tandem mass spectrometry.

endogenous UPF1. Northern blots of co-purified mRNAs indicated transcripts containing the RSE-Δ PTB sequence displayed substantially increased UPF1 association (*Figures 3B and C*). The inverse

relationship between PTBP1 and UPF1 association is consistent with our hypothesis that PTBP1 prevents UPF1 from associating with the 3'UTRs. Interestingly, although the level of UPF1 association on the transcripts containing RSE-ΔPTB+6xPTBBS is significantly reduced compared to RSE-ΔPTB, it was still higher than that of the wild-type RSE, indicating that the natural arrangement of the PTBP1 binding sites in the RSE may be optimal for antagonizing UPF1 binding. As a control to ensure that the PTBP1-mediated inhibition of UPF1 binding was due to cellular interactions rather than a consequence of protein-RNA reassortment in extracts, we further tested the ability of stably expressed tandem protein A-tagged UPF1 to co-purify mRNAs expressed in distinct cells (*Mili and Steitz, 2004*). In these experiments, we observed co-purification of mRNAs when tagged UPF1 and mRNAs were expressed in the same cells, but none when reporter RNA-containing extracts from cells lacking tagged protein were mixed with extracts containing tagged UPF1 (*Figure 3—figure supplement 1D, Figure 3—figure supplement 1E*).

## PTBP1 binding sites promote accumulation of RSV RNAs in chicken cells

As the physiological role of the RSE is the stabilization of unspliced RSV RNAs, we investigated the role of PTBP1 in stabilization of authentic viral RNAs (*Figure 4A*). To do so, we replaced the wt RSE sequence in the unspliced RSV RNA with the RSE-ΔPTB sequence or RSE-ΔPTB variants containing three or six artificial PTBP1 binding sites (RSE-ΔPTB+3xPTBBS or RSE-ΔPTB+6xPTBBS, respectively) and measured the steady state levels of the viral RNAs in chicken embryonic fibroblasts. In these experiments, the RSE-ΔPTB-containing viral RNAs were present at much lower levels than RNAs protected by the wt RSE sequence. Viral RNAs containing the RSE-ΔPTB+3xPTBBS sequence exhibited substantial rescue due to the inclusion of artificial PTB binding sites (*Figure 4B,C*), and six PTBP1 binding sites led to almost complete restoration of RNA levels, consistent with the effects of this RSE variant on PTBP1 and UPF1 binding (*Figure 3*).

## PTBP1 binding sites stabilize reporter transcripts in human cells

To test whether PTBP1 binding sites are sufficient to confer stability to NMD targets, we next inserted the wt RSE or the RSE-ΔPTB mutant sequence into reporter mRNAs containing the SMG5 3'UTR (*Figure 4D*). mRNA decay assays showed a dramatic UPF1-dependent reduction in the stability of the transcripts containing RSE-ΔPTB (half-life ~120 min) when compared to transcripts containing the wt RSE (half life >9 hr; *Figure 4E*; see *Figure 4—figure supplement 1* for UPF1 RNAi experiments). In fact, targeted mutation of the putative PTBP1 binding sites in the RSE resulted in transcript stability similar to that of mRNAs containing the inactive antisense RSE sequence. Transcripts containing the RSE-ΔPTB+6xPTBBS variant (half-life >9 hr) exhibited stability comparable to transcripts containing the wt RSE, indicating that the RSE regained most or all of its function after restoration of six PTB binding sites at arbitrarily chosen sites (*Figure 4E*).

## Introduction of PTBP1 binding sites to NMD-sensitive transcripts inhibits decay

The results presented above suggest that PTB binding is necessary for the protective activity of the RSE. To determine whether PTB binding downstream of a TC is sufficient to stabilize NMD targets, we constructed a series of reporter transcripts containing three or six model PTBP1 binding sites inserted at the 5' end of the NMD-inducing SMG5 3'UTR. In these experiments, we tested two arrangements of PTBP1 binding sites: separated by either 8 nt linker sequences (PTB3 and PTB6) or by 100 nt segments of the SMG5 3'UTR to better mimic the arrangement of PTB binding sites in the RSE (PTB3s and PTB6s; *Figure 5A*). In RNA decay assays, we found that addition of three PTBP1 binding sites significantly improved transcript stability, with the distributed binding sites having a marginally stronger effect (*Figure 5B*). Importantly, transcripts containing either arrangement of six PTBP1 binding sites were highly stabilized relative to control transcripts, demonstrating that PTBP1 binding is capable of recapitulating RSE function. Consistent with the position-dependence of the full-length RSE (*Figure 1—figure supplement 1*), mRNAs in which a cluster of six PTBP1 binding sites were introduced into the SMG5 3'UTR 600 nt or 1200 nt downstream of the TC were highly unstable (*Figure 5—figure supplement 1A,B*).

To further test the specificity of PTBP1's role as an inhibitor of NMD, we used a tethering approach to artificially target PTBP1 and other RNA binding proteins to sites immediately

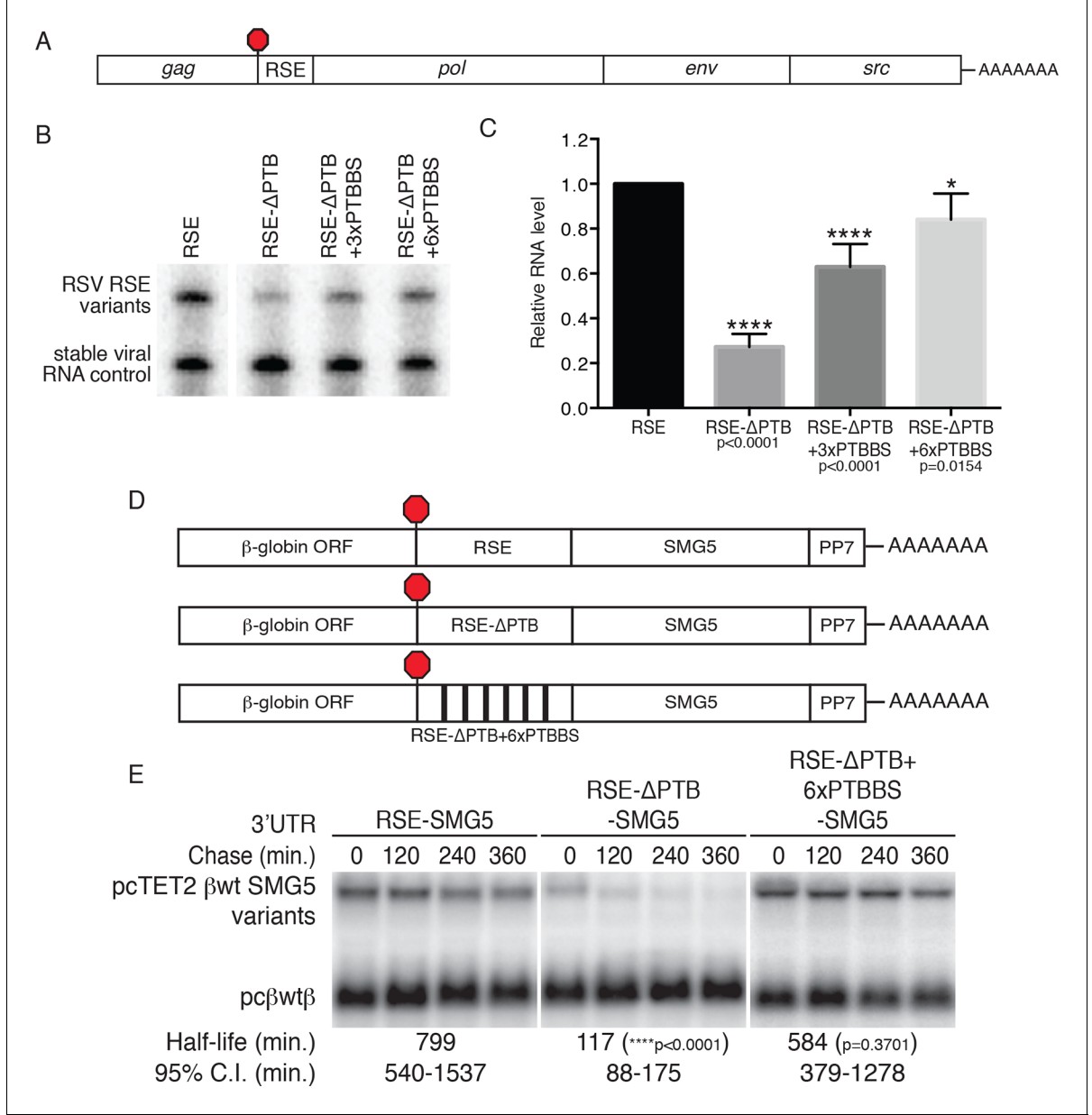

**Figure 4.** PTBP1 plays an essential role in the RSE's protective activity against NMD. (**A**) Schematic of unspliced RSV RNA expression constructs used for RNA accumulation assays in chicken fibroblasts. (**B**) RNase protection assays of RSV unspliced viral RNA containing the wt RSE or RSE variants. Experimental and control constructs were co-transfected into CEFs, and total cellular RNAs were harvested 43–48 hr post-transfection. Top band: protected fragment of the probe corresponding to the unspliced viral RNAs from the experimental constructs. Bottom band: stable viral loading control that protects a different sized fragment of the same probe due to a small in-frame deletion. (**C**) Quantification of RNase protection assays. Levels of the experimental unspliced RSV RNAs (top band) were normalized to levels of the transfection control (bottom band). RNA levels are reported as a fraction of RSV RNA containing wt RSE. Error bars indicate ± SD; n ≥ 5 (*p<0.05; ****p<0.0001 in two-tailed Student's t-tests when compared to RSE constructs). (**D**) Schematic of reporter mRNAs containing the β-globin gene, RSE variants (RSE, RSE-ΔPTB, or RSE-ΔPTB+6xPTBBS), and the full-length human SMG5 3'UTR. (**E**) Decay assays of the indicated reporter mRNAs. Tet-regulated transcripts (upper bands) were co-transfected with the constitutively expressed wild-type β globin reporter (pcβwtβ; bottom bands) in HeLa Tet-off cells. Levels of tet-regulated reporter mRNAs were normalized to levels of the wild-type β-globin transfection control. Half-lives and 95% confidence intervals were obtained from 3 independent experiments (p-values from two-tailed ANCOVA analyses when compared to pcTET2-βwt-SMG5). See also *Figure 4—figure supplement 1*.

The following figure supplement is available for figure 4:

**Figure supplement 1.** PTPB1 protects reporter mRNAs from NMD.

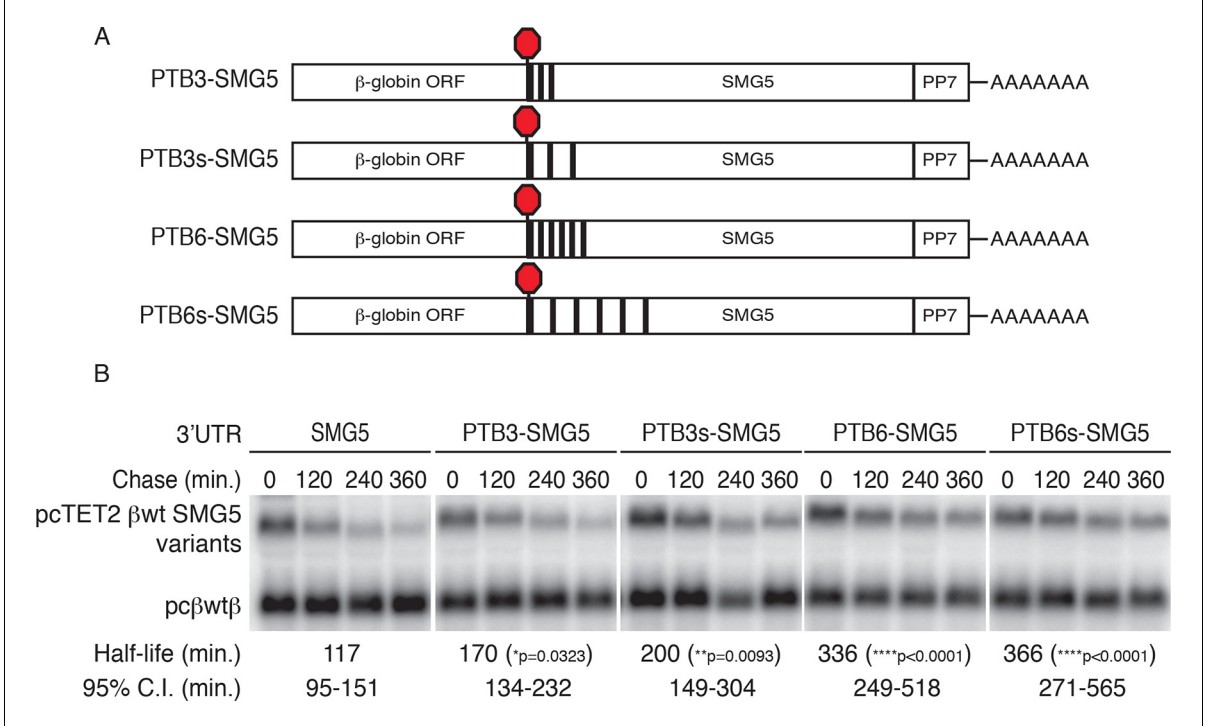

**Figure 5.** PTBP1 protects NMD-sensitive transcripts from NMD when artificially recruited to the 3'UTR. (**A**) Schematic of tet-regulated β-globin reporter mRNA constructs used in RNA decay assays. Three or six canonical PTBP1 binding sites were inserted into the SMG5 3'UTR. PTBP1 binding sites were inserted with short linker sequences between each site (PTB3 and PTB6), or spaced at 100 nt intervals within the SMG5 3'UTR (PTB3s and PTB6s). (**B**) Decay assays of reporter mRNAs containing PTBP1 binding sites. Constructs encoding the indicated tet-regulated transcripts (top bands) were co-transfected with the constitutively expressed wild-type β-globin reporter (pcβwtβ; bottom bands) in HeLa Tet-off cells. Levels of tet-regulated reporter mRNAs were normalized to levels of the wild-type β-globin transfection control. Half-lives and 95% confidence intervals were obtained from 3 independent experiments (p-values from two-tailed ANCOVA analyses when compared to pcTET2-βwt-SMG5). See also *Figure 5—figure supplement 1*.

The following figure supplement is available for figure 5:

**Figure supplement 1.** PTBP1 binding recapitulates the RSE's position-dependent NMD inhibition.

downstream from NMD-inducing stop codons. For these assays, we fused proteins of interest with the MS2 bacteriophage coat protein and co-expressed them with reporter mRNAs containing four MS2 hairpins placed at the 5' end of the SMG5 3'UTR. A PABPC1-MS2 coat protein fusion was included in this assay as a positive control, as PABPC1 has previously been reported to antagonize NMD when tethered to NMD-sensitive mRNAs (*Amrani et al., 2004*). Our results showed that the reporter mRNAs were comparably stabilized by co-expression with MS2-PABC1 or MS2-PTBP1, relative to co-expression with the MS2 coat protein alone (*Figure 5—figure supplement 1C,D*).

## PTBP1 protects many human transcripts with long 3'UTRs from NMD

To test the hypothesis that the mechanism used by the RSV viral RNA to evade NMD may be shared by mammalian mRNAs with long 3'UTRs, we performed RNA-seq analysis on 293 Tet-off cells transfected with anti-UPF1 and anti-PTBP1 siRNAs individually or in combination, using a validated non-silencing siRNA as a control. In biological replicates, mRNAs from 495 genes reproducibly decreased in abundance upon depletion of PTBP1, relative to their expression levels in cells treated with control siRNA (*Figure 6A*; see also *Figure 6—source data 1*), consistent with previous reports implicating PTBP1 in the stabilization of diverse human mRNAs. To determine whether the down-regulated mRNAs were made susceptible to NMD in the absence of PTBP1, we examined the effects of co-depletion of UPF1 and PTBP1. Of the transcripts down-regulated upon PTBP1 knockdown, we designated mRNAs from 188 genes observed to reproducibly increase in siPTB/siUPF1 vs siPTB alone as

'rescued.' On average, concurrent knockdown of UPF1 and PTBP1 caused an increase in mRNA levels among PTBP1-dependent genes (*Figure 6A*), while the average accumulation of PTBP1-dependent mRNAs was not altered upon UPF1 depletion alone (*Figure 6B*). Among all of the PTBP1-dependent genes, the extent of mRNA rescue upon co-depletion correlated with the reduction upon PTBP1 knockdown (*Figure 6A,C*). Consistent with a role for PTBP1 binding near the stop codon, this correlation was dependent on the presence of putative high-affinity PTBP1 hexamer binding sites in the region downstream of the TC (*Figure 6C*).

As 3'UTR length is a major determinant of NMD, we also asked whether the class of mRNAs protected from NMD by PTBP1 exhibited unusually long 3'UTRs. Indeed, mRNAs down-regulated in PTBP1-depleted cells had a median 3'UTR length of 1577 nt, significantly longer than the median 3'UTR length among all mRNAs expressed in the RNAseq dataset (1308 nt; *Figure 6D*; see Materials and methods for details). Most importantly, the population of mRNAs that were both significantly down-regulated by siPTBP1 treatment and rescued by co-depletion of PTBP1 and UPF1 exhibited an even greater median length of 1843 nt. Together, these data suggest that protection of human mRNAs containing long 3'UTRs by PTBP1 is widely exploited to minimize the impact of NMD on the transcriptome.

To verify the response of endogenous transcripts to depletion of PTBP1 and UPF1 alone and in combination, we selected a panel of transcripts for follow-up analysis by quantitative reverse transcription-PCR (qRT-PCR). These experiments confirmed that several mRNAs identified in RNA-seq analyses were down-regulated by PTBP1 depletion and fully or partially rescued by co-depletion of PTBP1 and UPF1 (*Figure 6E*). Importantly, PTBP1 mRNA levels were equal in the PTBP1 and PTBP1/UPF1 double knockdown conditions. Incomplete rescue of PTBP1-dependent mRNAs via UPF1 depletion could either be due to PTBP1 antagonizing multiple RNA decay pathways or due to insufficient UPF1 knockdown for complete abrogation of NMD activity. As alterations in RNA stability could be caused by changes in PTBP1-dependent alternative splicing events, we also examined the RNAseq data for evidence of altered splicing. With the exception of TPM1, a known target of PTBP1 splicing regulation, we did not observe differences in the splicing patterns of the selected genes (*Figure 6—figure supplement 1*).

Among nine genes validated by qRT-PCR as protected from NMD by PTB, eight have at least one putative PTB hexamer binding site within 150 nt of the annotated TC, and six of nine have two or more hexamer binding sites in that region (*Figure 6—figure supplement 2*) In addition, we analyzed previously published whole-genome CLIP data of PTBP1 in HeLa cells to probe a possible relationship between protection from NMD and PTBP1 binding in living cells (*Xue et al., 2013*). This analysis revealed PTBP1 CLIP reads mapping to the 3'UTRs of each of the validated mRNAs listed above (*Figure 6—figure supplement 2*; see below for further CLIP analysis).

## PTBP1 binding within 200 nt of the TC correlates with 3'UTR length and protection from NMD

In order to understand the global relationship between PTBP1 binding and NMD, we analyzed the RNA-seq data described above with respect to published PTBP1 (*Xue et al., 2013*) and UPF1 CLIP-seq datasets (*Zund et al., 2013*). As previously reported, analysis of aggregate binding of UPF1 to all mRNAs showed that UPF1 levels are low in coding sequences and increase downstream of TCs to reach a broad plateau throughout 3'UTRs (*Figure 7A*). In contrast, PTBP1 CLIP signal derived from exons was preferentially located within 200 nt of TCs, consistent with a role for PTBP1 in marking translation termination events.

Having observed enrichment of PTBP1 CLIP peaks near TCs, we next asked whether mRNAs with TC-proximal PTBP1 binding exhibited evidence of NMD evasion. In line with a role for PTBP1 in protecting potential NMD substrates, transcripts with PTBP1 CLIP peaks centered within the first 200 nt of the TC had longer 3'UTRs than those lacking peaks near the TC (*Figure 7B*). As a control, mRNAs with PTBP1 CLIP peaks centered between 200 and 500 nt from the TC exhibited no tendency to contain longer 3'UTRs than normal. Most importantly, mRNAs with PTBP1 peak centers within 200 nt of the TC were significantly less likely to be up-regulated upon UPF1 knockdown than mRNAs without TC-proximal peaks, indicating that this population of PTBP1-bound mRNAs is insensitive to NMD under normal conditions (*Figure 7C*). To avoid potential confounding effects of differences in 3'UTR lengths among distinct transcript classes, we used mRNAs with 3'UTRs falling between the 25th and 75th percentile of all 3'UTR lengths for these analyses (mRNAs with 3'UTRS from 484–2251

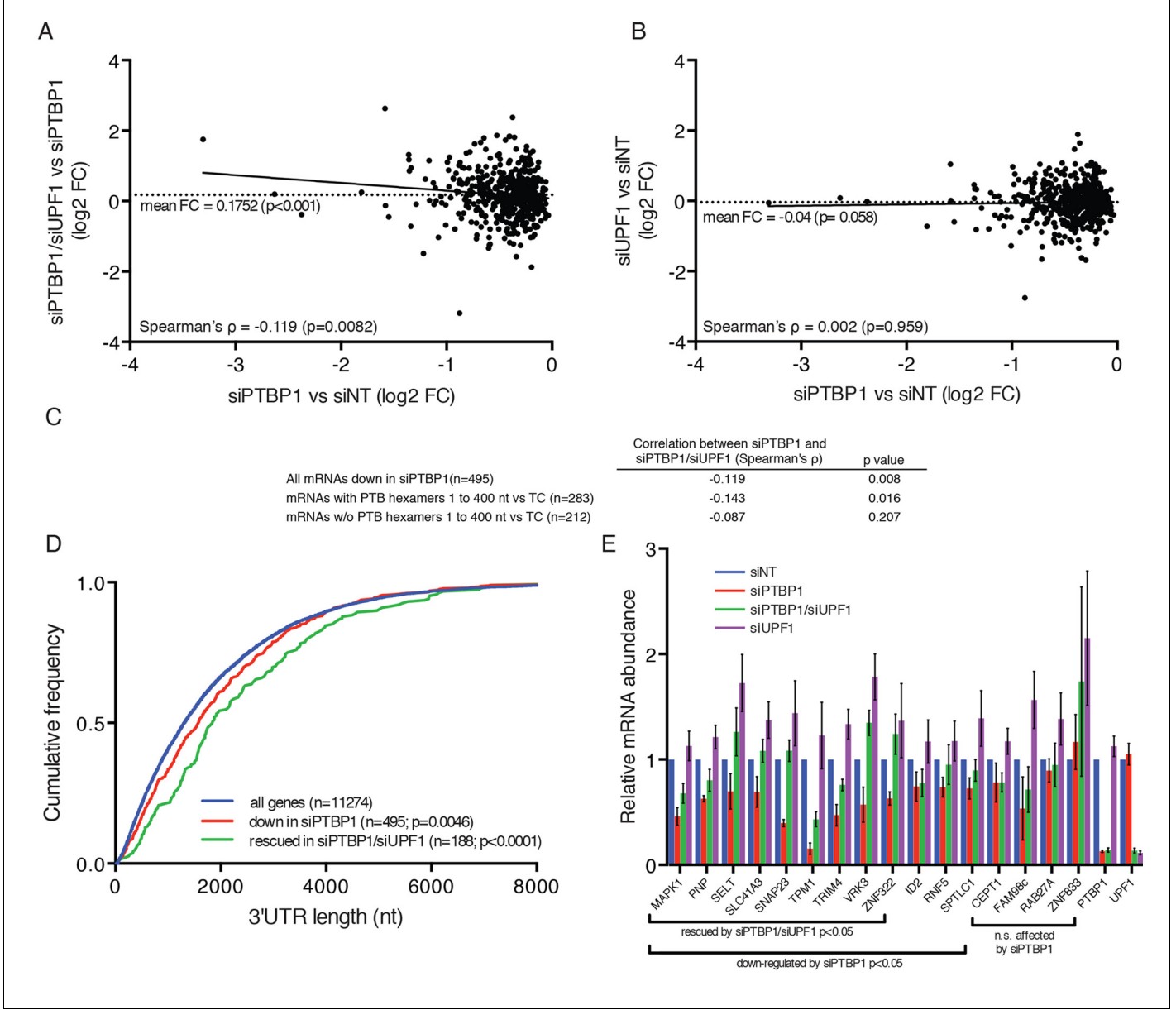

**Figure 6.** PTBP1 protects many human mRNAs with long 3'UTRs from NMD. (**A**) For mRNAs from 495 genes reproducibly down-regulated by PTBP1 depletion in RNAseq of biological replicates, fold change comparing siPTBP1/siUPF1 to siPTBP1 alone was plotted vs. fold change comparing siPTBP1 to siNT. Mean fold change among all analyzed genes (dashed line; p<0.0001 in two-tailed Student's t-test of null hypothesis of zero fold change), best fit determined by the least squares method (solid line) and correlation coefficient (Spearman's ρ) are indicated. See also *Figure 6—source data 1*. (**B**) Mean fold change in transcript abundance upon PTBP1 depletion versus siNT treatment was plotted against the mean fold change in abundance between siUPF1 and siNT conditions for the set of mRNAs as described in A. (**C**) Table of Spearman's correlation coefficients for all mRNAs reproducibly downregulated by siPTBP1, mRNAs containing one or more putative PTBP1 hexamer binding sites within 400 nt of the termination codon, and mRNAs lacking putative PTBP1 hexamers in that region. (**D**) mRNAs protected from NMD by PTBP1 have long 3'UTRs. Continuous distribution function (CDF) plot of annotated 3'UTR lengths among all expressed exemplar mRNAs (blue), mRNAs down-regulated by PTBP1 depletion (red), and mRNAs rescued by co-depletion of PTBP1 and UPF1 in RNA-seq analyses (green; see Materials and methods for details). Statistical significance was evaluated by two-tailed K-S test, comparing the indicated mRNA sets to the distribution of 3'UTR lengths among all exemplar mRNAs. (**E**) qRT-PCR analysis of mRNAs protected from NMD by PTBP1. Graph of average abundance of mRNAs normalized to housekeeping UBC control mRNAs (n=3, error bars indicate SD). Statistical significance was determined by two-tailed Student's t-test, comparing siPTBP1 to siNT and siPTBP1/siUPF1 to siPTBP1. See also *Figure 6—figure supplements 1*, *2* and *Figure 6—source data 1*.

The following source data and figure supplements are available for figure 6:

**Source data 1.** Table of mRNAs down-regulated by PTBP1 depletion.

*Figure 6 continued on next page*

*Figure 6 continued*

**Figure supplement 1.** PTBP1-mediated protection does not depend on changes in splicing.

**Figure supplement 2.** Analysis of PTBP1 binding to protected mRNAs.

nt). Providing further evidence of the importance of PTBP1 binding position relative to the terminating ribosome, mRNAs with PTBP1 CLIP peaks centered from 200 to 500 nt from the TC showed no significant trend toward protection from NMD (*Figure 7C*).

We further corroborated this analysis by examining the effect of UPF1 depletion on transcripts containing high-affinity PTBP1 binding sites near the TC. Using this approach, we found evidence that PTBP1 hexamer binding sites located in the immediate vicinity of the terminating ribosome are particularly important in determining protection from UPF1-mediated degradation. Specifically, transcripts containing PTBP1 hexamers within 50 nt of the TC were significantly less likely to be up-regulated upon UPF1 depletion than transcripts lacking putative binding sites in this interval, while transcripts with hexamers located 50–100 from the TC showed no apparent protection from NMD (*Figure 7D*). Together, these data suggest that human mRNAs have evolved to avoid recognition and degradation by the NMD machinery by recruiting PTBP1 to the vicinity of TCs.

## Discussion

Through a combination of functional and biochemical assays, we have shown that the RSE functions through PTBP1 to inhibit NMD. This activity is associated with a reduction in UPF1 binding to transcripts containing long 3′UTRs, an early event in the process of NMD. In our model, the RSE and PTBP1 effectively mask the true length of the downstream 3′UTR from UPF1, allowing the mRNA to elude surveillance. The ability of PTBP1 to antagonize UPF1 binding depends on its location on the mRNA; PTBP1 binding at the beginning of the 3′UTR causes a loss of UPF1 binding, while relocation to distal sites reduces this function. This position-dependence has the advantage of allowing stabilization in response to translation termination at the proper stop codon while preserving the ability of NMD to detect premature TCs. In contrast, a position-insensitive mechanism would prevent NMD from accurately detecting de novo nonsense mutations, reducing the scope of quality control.

In its role as a splicing regulator, PTBP1 functions in a partially redundant manner with its close paralogs, PTBP2 and PTBP3/ROD1 (*Keppetipola et al., 2012*). In many human cell types, including those used for this study, depletion of PTBP1 leads to induction of PTBP2, which compensates for a subset of PTBP1 functions. As it possesses highly similar biochemical activities, PTBP2 may also stabilize a subset of mRNAs usually protected by PTBP1, meaning that the population of mRNAs identified here as protected from NMD is likely to underestimate the role of PTB paralogs in NMD inhibition. In addition, we expect that marking genuine TCs with PTBP1 is just one strategy employed by long 3′UTRs to evade NMD. Supporting the existence of diverse solutions to the problem, yeast Pub1p was found to stabilize certain NMD targets by means that remain undefined (*Ruiz-Echevarria and Peltz, 2000*), and manipulation of 3′UTR structure to bring the poly-A tail in spatial proximity to the TC has been shown to stabilize reporter mRNAs (*Eberle et al., 2008*).

Extensive work on the role of PTB proteins in mediating exon exclusion in pre-mRNA splicing provides a template for a proposed mechanism for PTBP1's anti-UPF1 activity. PTBP1 contains four RRMs, each with affinity for CU-containing sequences (*Oberstrass et al., 2005*). Of these, RRM3 and RRM4 are oriented back-to-back, resulting in RNA loops of 15 nt or more between sites of PTBP1 contact (*Lamichhane et al., 2010*). These short-range looping events may also be accompanied by capture of distant CU-rich sequences by individual RRMs forming larger looped RNA structures (*Clerte and Hall, 2009*; *Kafasla et al., 2009*). It has been suggested that long-range interactions involving multiple PTBP1 molcules bound to a single RNA may be responsible for excluding binding of splicing factors, resulting in splicing repression (*Chou et al., 2000*; *Amir-Ahmady et al., 2005*; *Cherny et al., 2010*; *Lamichhane et al., 2010*; *Sharma et al., 2011*). Alternatively, PTBP1 may interact with proteins such as MATRIN 3 to promote the assembly of higher-order repressive complexes (*Joshi et al., 2011*). By direct analogy to its role in splicing, PTBP1 assembled on multiple TC-

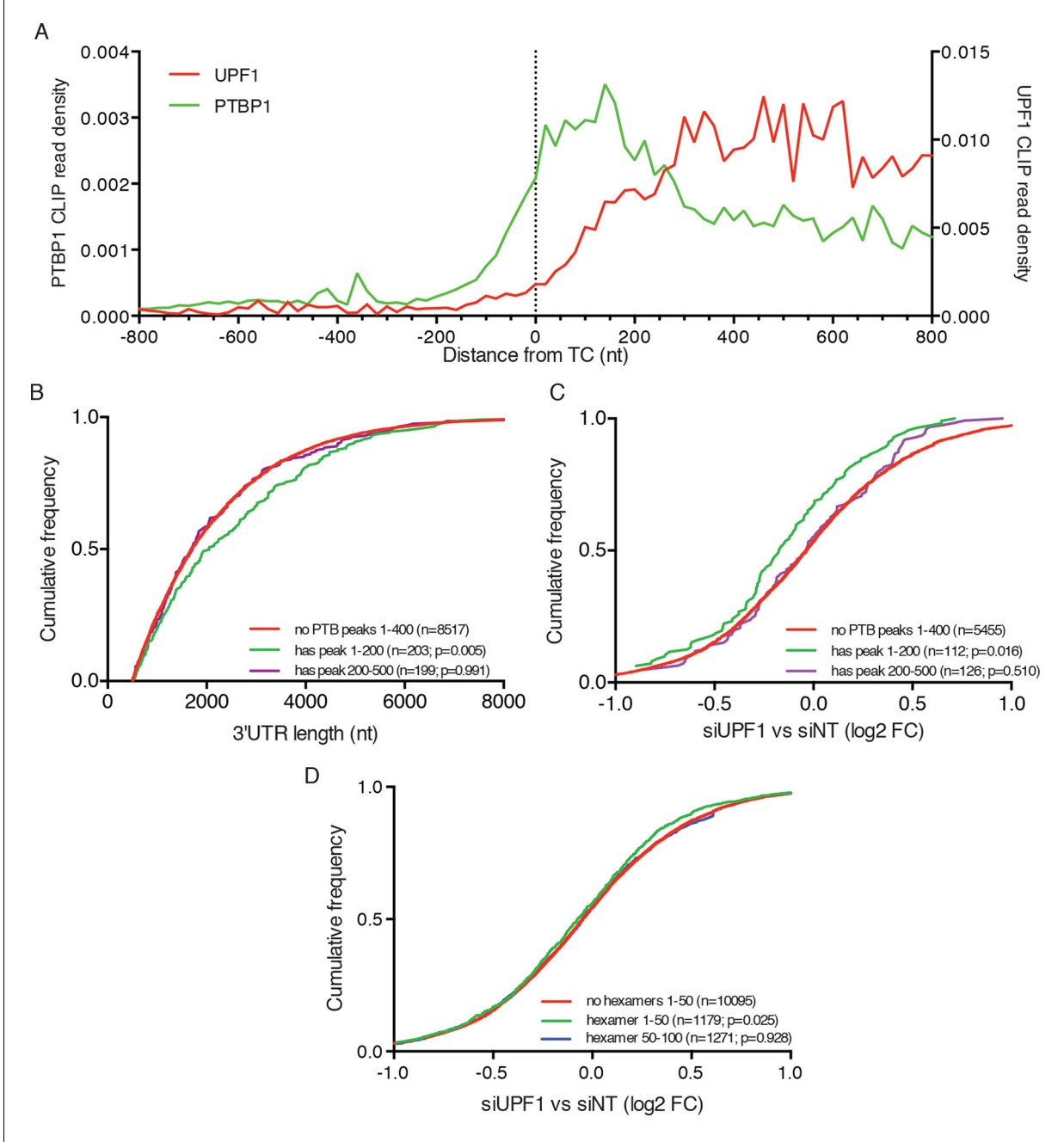

**Figure 7.** PTBP1 binding near stop codons is correlated with 3'UTR length and NMD evasion. (**A**) Density of PTBP1 (green, left axis) and UPF1 (red, right axis) CLIP reads derived from peaks called using PIPE-CLIP software, plotted relative to TC position. A bin size of 20 nt was used to determine read density; bin sizes ranging from 5 nt to 20 nt and plots of peak occurrences showed similar patterns. (**B**) CDF plot of annotated 3'UTR lengths among mRNAs containing or lacking PTBP1 CLIP peaks centered in the indicated intervals relative to the TC. P-values were calculated in two-tailed K-S tests of 3'UTR lengths in the indicated mRNA classes compared to mRNAs lacking PTBP1 peaks. Only mRNAs with 3'UTRs greater than 500 nt were included to avoid bias due to selection of transcripts containing CLIP peaks. (**C**) CDF plot of log2 fold changes in mRNA abundance in UPF1 siRNA vs siNT RNAseq. mRNAs with 3'UTR lengths in the middle 50% of the overall distribution (484–2251 nt) were selected to avoid confounding effects of increased 3'UTR lengths among mRNAs with PTBP1 peaks. P-values were determined by two-tailed K-S tests, comparing the indicated mRNA classes to mRNAs lacking PTBP1 peaks from +1–400 nt. (**D**) CDF plot as in C. mRNAs were classified according to the presence of one or more predicted high-affinity PTBP1 hexamer binding sites in the indicated intervals relative to annotated TCs (see Materials and methods for details; the top 6 hexamers identified in previous PTBP1 CLIP seq, associated with >50% of observed CLIP peaks, were used for this analysis; *Xue et al., 2009*). Statistical significance was determined by two-tailed K-S tests comparing mRNAs with putative PTBP1 binding sites in the indicated positions to mRNAs lacking hexamer binding sites at positions from 1 to 50 nt downstream of the TC.

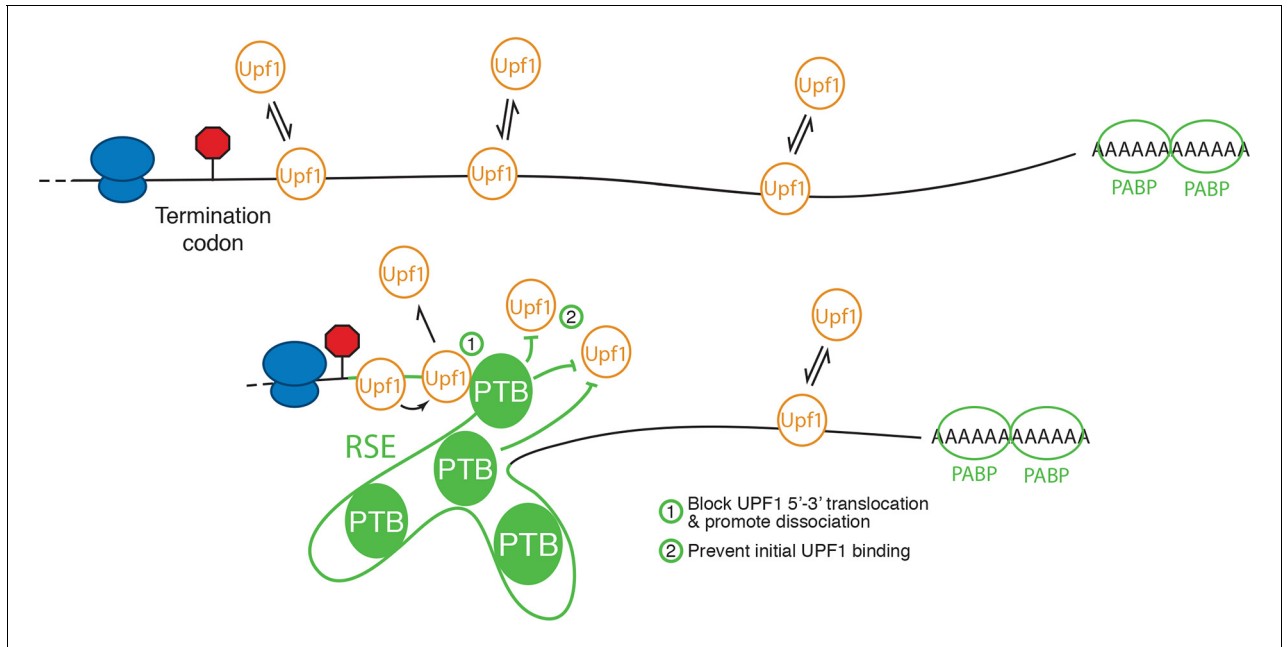

**Figure 8.** Model for NMD inhibition by PTBP1. In the absence of the RSE or other sequences capable of recruiting PTBP1, UPF1 binds 3'UTRs in a length-dependent manner, potentiating NMD (top). PTBP1 can bind the RSE at multiple sites throughout the 400 nt sequence. This establishes an mRNP structure in the vicinity of the stop codon that is refractory to UPF1 binding, possibly by (1) blocking 5'-3' translocation of UPF1 or (2) preventing initial binding of UPF1 to the mRNA. The NMD pathway thus judges the 3'UTR to be short and fails to degrade the mRNA.

proximal CU-rich sequences could thus result in a zone of UPF1 exclusion, preventing the initial assembly of UPF1 into mRNPs (*Figure 8*). In addition to preventing initial UPF1 association with 3'UTRs, PTBP1 could affect UPF1 already bound to an mRNA. In this scenario, PTBP1 may, as has been recently described for cytoplasmic poly-A binding protein, stimulate ATPase-dependent dissociation of UPF1 from the mRNP (*Lee et al., 2015*).

The PTB family of proteins is a pioneering example of the coupling of alternative splicing regulation with NMD. A series of studies has elucidated a complex regulatory network in which PTBP1 and PTBP2 auto- and cross-regulate each other to favor production of PTC-containing transcripts that are efficiently degraded by NMD (*Markovtsov et al., 2000*; *Romanelli et al., 2000*, *2005*; *Rahman et al., 2002*; *Wollerton et al., 2004*; *Boutz et al., 2007*; *Spellman et al., 2007*). In most tissues, PTBP1 regulates the alternative splicing of PTBP2 to favor production of PTC-containing transcripts that are efficiently degraded by NMD. In the brain and testes, PTBP1 is down-regulated, allowing increased expression of PTBP2 and execution of tissue-specific splicing programs. As in their roles in alternative splicing, we expect that PTBP1 and PTBP2 may stabilize partially overlapping sets of potential NMD substrates in distinct cell types (*Keppetipola et al., 2012*).

## Materials and methods

### Plasmids and Oligonucleotides

Wild-type RSE, RSE mutants, anti-sense RSE, and SMG5-397 sequences were amplified by PCR from the E/S wild-type RSV vector (*Withers and Beemon, 2011*), pcTET2-βwt-SMG5 (Jens Lykke-Andersen, UCSD) or synthesized (Integrated DNA Technologies, Coralville, IA) and inserted into reporter mRNAs at the *Not*I site of pcTET2-βwt-SMG5, pcTET2-βwt-GAPDH, or pcTET2-βwt-GAPDH-AdML or the *Bam*HI (5') or *Pme*I (3') sites of pcDNATPEGFP-GAP (*Hogg and Goff, 2010*; *Singh et al., 2008*). To create pcTET2-βwt-PTB3-SMG5 and pcTET2-βwt-PTB6-SMG5 or pcTET2-βwt-4xMS2-SMG5, DNA cassettes containing three or six PTBP1 binding sites (CTCTCTCTTCTTCTT and TCTTC-TTCTTCTTCT; as used in *Xue et al., 2009*) or four MS2 binding sites, respectively, were introduced into the *Not*I site in the parental pcTET2-βwt-SMG5 plasmid. For pcTET2-βwt-MPTB6-SMG5, an

*Age*I site was engineered into the SMG5 sequence of pcTET2-βwt-SMG5 600 nt downstream of the TC, and DNA cassettes containing six PTBP1 binding sites were introduced into the *Age*I site. To construct pcTET2-βwt-3'PTB6-SMG5, DNA cassettes containing six PTBP1 binding sites were introduced into the *Xba*I site in the parental pcTET2-βwt-SMG5 plasmid, 1343 nt downstream of the TC. In pcTET2-βwt-PTB3s-SMG5 and pcTET2-βwt-PTB6s-SMG5, DNA fragments containing the PTBP1 binding sites at desired locations were synthesized (Life Technologies, Frederick, MD) and cloned into pcTET2-βwt-SMG5 between *Not*I and *Age*I sites. To generate RSV RSE variants, RSE-ΔPTB and RSE-ΔPTB+3xPTBBS sequences were amplified by PCR from the respective pcTET2-βwt-SMG5 variant plasmids and cloned into the E/S wild-type RSV vector between *Eag*I and *Spe*I sites to replace the wild-type RSE sequence (*Withers and Beemon, 2011*). For stable expression of tagged UPF1, pBABE hygro (*Morgenstern and Land, 1990*) was modified to contain tandem protein A domains and a TEV protease cleavage site between the *Nae*I and *Not*I sites, and full-length UPF1 coding sequences was inserted between the *Not*I and *Xho*I sites. To construct plasmids expressing PTBP1 or PABPC1 with the MS2 coat protein fused at the N-terminus, the MS2cp V75E; A81G mutant (*LeCuyer et al., 1995*) was amplified by PCR and inserted into the *Not*I site of pQCXIP (Clontech, Mountain View, CA). DNA fragments encoding PTBP1 or PABC1 were amplified from respective cDNAs (Open Biosystems/Dharmacon, Lafayette, CO) and cloned into pCQXIP-MS2CP plasmid between *Bam*HI and *Xho*I sites. Sequences of siRNAs (Thermo Scientific, Philadelphia, PA; Dharmacon, Lafayette, CO) used are as follows: UPF1, GAUGCAGUUCCGCUCCAUUUU; PTPB1, CUUCCAUCAUUCCAGAGAAUU; non-targeting, UAAGGCUAUGAAGAGAUAC (*Mendell et al., 2004*; *Wagner and Garcia-Blanco, 2002*).

## Cell culture and extracts for mRNP purification and immunoprecipitation

HEK-293T cells maintained in DMEM supplemented with 10% fetal bovine serum, 100U of penicillin/ streptomycin, and 0.3 mg/ml L-Glutamine (Life Technologies) were used for mRNP purification experiments. Cells were transfected with desired constructs by calcium phosphate as described (*Hogg and Collins, 2007*). Cells were harvested 48 hr post transfection, resuspended in hypotonic lysis buffer (HLB; 20 mM HEPES pH 7.6, 2 mM MgCl$_2$, 10% glycerol, 1 mM DTT, supplemented with protease inhibitors), subjected to freeze-thaw lysis, and extracted in 150 mM NaCl as described (*Hogg and Collins, 2007*).

## Immunoprecipitation and mRNP purification

PP7-based mRNP purification and immunoprecipitation were performed as described, with minor modifications (*Hogg and Goff, 2010*). For immunoprecipitation, goat anti-Upf1 antibody (Santa Cruz), goat anti-PTBP1 antibody (Abcam, Cambridge, MA), or nonspecific goat IgG (Sigma, St. Louis, MO) pre-bound to protein G Dynabeads (Life Technologies) were incubated in extracts for 1 hr, followed by extensive washing with HLB supplemented with 150 mM NaCl and 0.1% NP-40. Bound RNA was eluted from beads in Trizol reagent (Life Technologies). Where indicated, cells were treated with 100 µg/ml puromycin (Sigma, St. Louis, MO) for 4 hr prior to cell extract preparation. To quantify the relative recovery of RNA, the amount of RNA in the bound fraction was divided by the amount in the total fraction for experimental and recovery control RNAs, and the recovery of the experimental RNA was normalized using the recovery of the co-expressed control RNA. Average and SD of relative mRNA recoveries from biological replicates performed using extracts from cells transfected separately are reported.

Experiments to test for post-lysis rearrangement of protein-RNA interactions were conducted as described above, except that parental 293 Tet-off cells or cells stably transduced with retroviral vectors encoding tandem protein A-tagged UPF1 were transfected with mRNA expression vectors. Equal amounts of extract (determined by total protein content) prepared from cells expressing both tagged protein and exogenous mRNA were mixed with extracts from parental cells lacking exogenous RNA expression, or extracts from parental cells expressing only exogenous mRNAs were mixed with extracts from cells expressing only tagged UPF1. Affinity purifications were carried out using rabbit IgG conjugated with M-270 epoxy Dynabeads (Life Technologies) according to the manufacturer's instructions.

For mRNP purification, whole cell extracts of HEK293T cells were adjusted to 5 mg/mL protein in HLB with 150 mM NaCl. Purified ZZ-tev-PP7CP (0.5 µg) was mixed with 500 µl of extracts by rotating for 60 min at 4°C. 50 µl of the Rabbit IgG-conjugated M-270 Dynabeads were added to 500 µl of extracts premixed with ZZ-tev-PP7CP. The mixture was rotated for an additional 60 min at 4°C to allow binding. The beads were then washed extensively in HLB supplemented with 150 mM NaCl, 0.1% NP-40, and 1 mM DTT. Bound RNA and protein were eluted from beads in LDS sample buffer (Life Technologies) or in HLB supplemented with 1 M NaCl (in preparation for mass spectrometry).

## Mass spectrometry

Proteins eluted from mRNP purifications were precipitated in cold acetone (Thermo Scientific) and subsequently resuspended in 6 M urea (Thermo Scientific). Proteins were reduced in 20 mM DTT (Thermo Scientific), alkylated in 50 mM iodoacetamide (Thermo Scientific), diluted to 2 M urea, and digested with trypsin (Promega, Madison, WI) overnight. Peptides were purified using $C_{18}$ Zip-Tips (EMD Millipore, Taunton, MA), according to the manufacturer's instructions. Samples were loaded onto a Zorbax $C_{18}$ trap column (Agilent Tech, Santa Clara, CA) using an on-line Eksigent (Dublin, CA) nano-LC ultra HPLC system to further desalt the peptide mixture. Peptides were separated on a 10 cm Picofrit $C_{18}$ column (New Objective, Woburn, MA) using a 90 min linear gradient of 5–35% acetonitrile/water containing 0.1% formic acid. The samples were ionized via electrospray ionization (ESI) in positive mode and analyzed on a LTQ Orbitrap Velos mass spectrometer (Thermo Electron, San Jose, CA). LC MS/MS experiments were carried out using a 'data-dependent' analysis in which the top 6 most intense precursor ions from the full MS precursor scan were selected for fragmentation via collision-induced dissociation (CID). Precursor ions were measured in the orbitrap at a resolution of 30,000 (m/z = 400) and all fragment ions were analyzed in the linear ion-trap.

All LC MS/MS data was searched against the human Swissprot database (20312 sequences) using MASCOT to obtain peptide and protein identifications. Mass tolerances of ± 20 ppm and ± 0.8 Da were selected for precursor and fragment ions, respectively. Trypsin was specified as the digestion enzyme allowing for up to 2 missed cleavages. Carbamidomethylation (C) was selected as a static modification and oxidation (M) was selected as a variable modification. MASCOT database search results were then imported into Scaffold (Proteome Software, Portland, OR) for further validation of peptide and protein identifications.

## mRNA decay assays

HeLa Tet-off Advance cells (Clontech, Mountain View, CA) maintained in DMEM supplemented with 10% FBS and 5 ng/ml doxycycline (Sigma, St. Louis, MO) were transfected in 60 mm plates using Turbofect (Thermo Scientific), according to the manufacturer's instructions. For each plate, 800 ng of the indicated pcTET2-βwt plasmid was co-transfected with 200 ng pcβwtβ and 1000 ng pcDNA3.1 empty vector (*Lykke-Andersen et al., 2000*; *Singh et al., 2008*). Cells were split into four equal aliquots in 12-well plates 24 hr post-transfection. The next day, cells were washed with 1 ml PBS and incubated in medium without doxycycline for 4 hr. Transcription was quenched by adding doxycycline to a final concentration of 1 µg/ml, and cells were harvested in Trizol (Life Technologies) after 30 min and at the indicated intervals. mRNA levels from biological replicates performed using extracts from cells transfected separately are reported for all decay assays. Prism software (Graph-Pad) was used to graph the fraction of remaining RNA at the indicated time points on semi-log plots, obtain the rate of RNA decay ($k_{decay}$) by fitting to a linear equation using the least squares method, determine 95% confidence intervals, and test the significance of differences among the slopes (two-tailed ANCOVA). The half-lives of the mRNAs were calculated using the equation: $t_{1/2} = \ln2/k_{decay}$.

To conduct mRNA decay assays in cells depleted for specific factors by RNAi, 293 Tet-off cells were first transfected with 120 pmol (single siRNA treatment) or 60 pmol (dual siRNA treatment) of each of the indicated siRNAs (Thermo Scientific; Dharmacon, Lafayette, CO) in 6-well plates using Lipofectamine RNAiMAX (Life Technologies), according to the manufacturer's instructions. Cells were split into four equal aliquots in 24-well plates 24 hr post-transfection. The next day, cells were transfected with reporter constructs using Turbofect (Thermo Scientific). For each well, 100 ng of the indicated pcTET2-βwt plasmid was co-transfected with 25 ng pcβwtβ and 125 ng pcDNA3.1 empty vector, and mRNA decay assays were performed and analyzed as described above.

## Northern blotting

RNA was isolated using Trizol (Life Technologies) and resolved on formaldehyde/agarose gels. [32]P-labeled in vitro transcribed probes against β-globin or random-primed DNA probes against GFP were used for detection of mRNAs. Northern blots were imaged on Storm or Typhoon Trio scanners and quantification was performed using ImageQuant software (GE Healthcare, Pittsburgh, PA).

## RNase protection assays

Secondary chicken embryo fibroblast (CEF) cultures were grown at 39°C and 5% $CO_2$ in medium 199 supplemented with 2% tryptose phosphate broth, 1% chick serum, 1% calf serum and 1% antibiotic-antimycotic. Transient transfections were performed as described (*Paca et al., 2000*), using 100 μg/mL of DEAE-dextran in serum-free medium 199. Cells were transfected with 3 μg of DNA in 6 cm dishes that were 80% confluent. Probe generation and RNase protection assays were performed as previously described (*LeBlanc and Beemon, 2004*). Total cellular RNA was harvested from CEFs 43–48 hr post-transfection using RNA-Bee (Tel-test). RNAs were hybridized overnight with ~250,000 cpm of [α-[32]P] GTP radiolabeled probe, then digested with 10 U/mL of RNase T1 and 5 μg/mL of RNase A at 33°C for 45 min. Sodium dodecyl sulfate and proteinase K were added and samples were incubated at 37°C for 20 min to halt digestion. RNAs were extracted with phenol-chloroform-isoamyl alcohol (25:24:1), followed by ethanol precipitation. RNAs were resuspended in 95% form-amide loading dye and denatured for 3 min at 95°C. Samples were electrophoresed on 6% acrylam-ide-8 M urea sequencing gels and RNA levels were quantified using a Typhoon Phosphorimager (GE Healthcare) and ImageQuant software (GE Healthcare). mRNA levels from biological replicates per-formed using extracts from cells transfected separately are reported.

## qPCR

293 Tet-off cells were transfected with siRNAs in 6-well plates using RNAiMAX (Life Technologies), as described above. Cells were harvested 72 hr post-transfection and total RNA was extracted using RNeasy Mini Kit (Qiagen, Venlo, Netherlands). 500 ng of each total RNA sample was used for cDNA synthesis using the Maxima First Strand cDNA Synthesis Kit for RT-qPCR (Thermo Scientific). cDNAs were diluted 1:40 with water and used for qPCR with the FastStart Essential DNA Green Master kit on a LightCycler 96 thermocycler (Roche, Basel, Switzerland). Relative transcript abundances were calculated by the ΔΔCt method, and statistical significance was assessed by two-tailed Student's t-test (Prism software, GraphPad). mRNA levels from biological replicates performed using extracts from cells transfected separately are reported.

## RNA-seq

For RNA-seq, 1 μg of each total RNA sample was used for library preparation. Ribosomal RNAs (rRNAs) were subtracted from total RNA with Ribo-Zero rRNA Removal Kits (Human/Mouse/Rat; Epi-centre). Following purification, RNA was fragmented and converted into first strand cDNA using reverse transcriptase and random primers, followed by second strand cDNA synthesis using DNA Polymerase I and RNase H. Following adapter ligation, products were purified and amplified by PCR to create final cDNA libraries. cDNA libraries were validated using an Illumina Miseq and paired-end 50 bp sequencing was performed on an Illumina HiSeq 2000 instrument. Raw data in FASTQ format was aligned to the reference genome hg19 using TopHat2/bowtie2. Reads mapping to the 3'UTRs of RefSeq genes were counted by Bedtools. Every RefSeq transcript and its 3'UTR expression level were normalized by three steps: calculating raw reads into RPKM (reads per-kilobase-per million), 75th percentile normalization and log2 transformation. The dataset was filtered to remove mRNAs represented by fewer than 25 3'UTR-mapped reads, and differentially expressed genes were selected by one-way ANOVA analysis (*Figure 6*; $p < 0.05$). The RefSeq transcript with the longest annotated 3'UTR for each gene was used for construction of the reference set of mRNAs used in all subsequent analyses (designated 'exemplar' RefSeq mRNAs). For *Figures 6 and 7*, the six most highly enriched hexamers in PTBP1 CLIP experiments (*Xue et al., 2009*; UUCUCU, UCUUCU, UCU-CUU, UCUCUG, CUUUCU, and CUUCUC), representing over 50% of all PTBP1 CLIP peaks, were used.

## CLIP analysis

PTBP1 (GSE42701; *Xue et al., 2013*) and UPF1 (GSE47976; *Zund et al., 2013*) CLIP datasets were obtained from the NCBI SRA database. For UPF1 data, reads not starting with GTT (derived from the iCLIP RT primer) were removed to avoid ambiguous reads, and duplicate reads with identical degenerate barcode sequences were removed with fastx_collapser (http://hannonlab.cshl.edu/fastx_toolkit/commandline.html). The first 7 nucleotides (containing barcodes) of collapsed reads were trimmed before alignment. For PTBP1, sequence duplicates were collapsed, and the first three nucleotides were trimmed before alignment. Novoalign (Novocraft), which allows the detection of small insertions, deletions and substitutions, was used to map CLIP reads to the hg19 human genome. Basic parameters were the same as those previously described (*Moore et al., 2014*): a maximum of two-nucleotide mismatches and a minimum of 25 high quality matches were allowed. Bam files were then input into PIPE-CLIP (*Chen et al., 2014*) to identify peaks with default parameters for iCLIP (UPF1) and HITS-CLIP (PTBP1). The peaks were annotated with Pavis (*Huang et al., 2013*), and HOMER (*Heinz et al., 2010*) was used to quantify CLIP tag density around stop codons. Only peaks narrower than 500 nucleotides were used for further analysis. Stop codon chromosomal locations were retrieved from the refGene table in hg19 of UCSC. For *Figure 7B*, only mRNAs with 3'UTRs greater than 500 nt were analyzed to avoid bias due to the selection of transcripts containing CLIP peaks in the indicated intervals.

## Sequences

Sequences used for RSE wild-type and mutant constructs are listed below, with mutated putative PTBP1 binding sites underlined.

### WT RSE sequence

AGGGCCA<u>CTGTTCTC</u>ACTGTTGCGCTACATCTGGCTA<u>TTCCGCTC</u>AAATGGAAGCCAGACCAC ACGCCTGTGTGGATTGACCAGTGG<u>CCCCTCCCT</u>GAAGGTAAACTTGTAGCGCTAACGCAATTAG TGGAAAAAGAATTACAGTTAGGACATATAGAA<u>CCTTC</u>ACTTAGTTGTTGGAACACA<u>CCTGTCTTC</u> GTGATCCGGAAGG<u>CTTCC</u>GGGTCTTACCGCTTACTGCATGATTTGCGCGCTGTTAACGCCAAG <u>CTTGTTCCTTTT</u>GGGGCCGTCCAACAGGGGGCGCCAG<u>TTCTCTCC</u>GCGCTCCCGCGTGGCTGG CCCCTGATGGTCTTAGACCTCAAGGATTG<u>CTTCTTTTC</u>TAT<u>CCCTCTT</u>GCGGAACAAGATCGCG AAG<u>CTTTTGC</u>ATTTA

### RSE-ΔPTB sequence

AGGGCCA<u>CTGTGATC</u>AAGGTTGCGCTACATCTGGAGA<u>TGACGAGC</u>AAATGGAAGCCAGACC ACACGCCTGTGTGGATTGACCAGTGG<u>CCCAGCCAGG</u>AAGGTAAAATGGTAGCGCTAACGCAATT AGTGGAAAAAGAATTACAGTTAGGACATATAGA<u>AACAGT</u>CAAGTAGTTGTTGGAACACA <u>CAGGTCTGA</u>GTGATCCGGAAGG<u>CTGAC</u>GGGTAGTACCGCTGAATGCATGATTTGCGCGCTGTT AACGCCAAG<u>AGTGTGACTTGT</u>GGGGCCGTCCAACAGGGGGCGCCAG<u>TTAGCGCA</u>GCGCTACCG CGTGGCTGGCACCTGATGGGATTAGACCGAAAGGATTG<u>CTGATGTTA</u>TATCCAGCTAGCGGAAC AAGATCGCGAAG<u>CTGTTGC</u>ATTTA

## Acknowledgements

We thank Jens Lykke-Andersen and Johanna Withers for generously providing reagents and Stephen P Goff, Lisa Postow, and members of the Hogg and Beemon labs for insightful comments. Mass spectrometry experiments and analysis were performed by Stephen Swatkoski and Marjan Gucek in the NHLBI Proteomics Core, and high-throughput sequencing was conducted and analyzed by Jun Zhu, Poching Liu, and Yanqin Yang in the NHLBI DNA Sequencing and Computational Biology Core and Eric Billings, Shouguo Gao, and Xujing Wang in the NHLBI Bioinformatics and Systems Biology Core.

## Additional information

### Funding

| Funder | Grant reference number | Author |
|---|---|---|
| National Institute of General Medical Sciences | P50 GM103297 | Karen L Beemon |
| National Cancer Institute | RO1 CA048746 | Karen L Beemon |
| National Heart, Lung, and Blood Institute | Intramural Research Program | J Robert Hogg |

The funders had no role in study design, data collection and interpretation, or the decision to submit the work for publication.

### Author contributions

ZG, JRH, Conception and design, Acquisition of data, Analysis and interpretation of data, Drafting or revising the article; BLQ, Conception and design, Acquisition of data, Analysis and interpretation of data; KLB, Conception and design, Analysis and interpretation of data

### Author ORCIDs

Bao Lin Quek, http://orcid.org/0000-0001-7306-3908
J Robert Hogg, http://orcid.org/0000-0001-5729-5135

## Additional files

### Major datasets

The following datasets were generated:

| Author(s) | Year | Dataset title | Dataset URL | Database, license, and accessibility information |
|---|---|---|---|---|
| Ge Z, Quek B, Beemon KL, Hogg JR | 2016 | Polypyrimidine tract binding protein 1 protects mRNAs from recognition by the nonsense-mediated mRNA decay pathway | http://www.ncbi.nlm.nih.gov/geo/query/acc.cgi?token=odszwkmyjdabnop&acc=GSE59884 | Publicly available at the NCBI Gene Expression Omnibus (Accession no: GSE59884). |

The following previously published datasets were used:

| Author(s) | Year | Dataset title | Dataset URL | Database, license, and accessibility information |
|---|---|---|---|---|
| Zünd D, Gruber AR, Zavolan M, Mühlemann O | 2013 | Translation-dependent displacement of UPF1 from coding sequences causes its enrichment in 3' UTRs | http://www.ncbi.nlm.nih.gov/geo/query/acc.cgi?acc=GSE47976 | Publicly available at the NCBI Gene Expression Omnibus (Accession no: GSE47976). |
| Xue Y, Ouyang K, Huang J, Zhou Y, Ouyang H, Wang G, Wu Q, Wei C, Bi Y, Jiang L, Cai Z, Sun H, Zhang K, Zhang Y, Chen J, Fu X | 2013 | Direct Conversion of Fibroblasts to Neurons by Reprogramming PTB-Regulated microRNA Circuits | http://www.ncbi.nlm.nih.gov/geo/query/acc.cgi?acc=GSE42701 | Publicly available at the NCBI Gene Expression Omnibus (Accession no: GSE42701). |

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
