## [Decision Letter]

Thank you for submitting your work entitled "PTBP1 excludes UPF1 to inhibit nonsense-mediated mRNA decay" for consideration by *eLife*. Your article has been reviewed by three peer reviewers, one of whom is a member of our Board of Reviewing Editors, and the evaluation has been overseen by Detlef Weigel as the Senior Editor.

The reviewers have discussed the reviews with one another and the Reviewing Editor has drafted this decision to help you prepare a revised submission.

Summary:

In this study, the Hogg lab reports a role for the hnRNP protein, PTBP1, in binding to long 3'UTRs of target mRNAs leading to the exclusion of UPF1 and resulting in protection from the NMD response. This is an original and well-written manuscript that tests an interesting hypothesis and identifies PTBP1 as an NMD suppressor. The referees asked for the following experiments to more strongly support the conclusions.

Essential revisions:

1) The effects are relatively weak. One way to strengthen the conclusions will be to insert PTBP1 binding sites downstream of nonsense codons in well characterized NMD reporters (e.g., β-globin or TCR-β), that are efficiently degraded by NMD or to tether PTBP1 downstream of a stop codon.

2) The evidence for the competition for mRNA binding model relies on the RNA-IP data. However, no X-linking was involved in the protocol and the authors cannot rule out about post-lysis rearrangements possibly confounding the results (as described by Mili and Steitz). The authors should perform control experiments to test the extent of such rearrangements in their assays.

3) To use the GFP-bGH construct for normalization seems inadequate, since the authors mention that this binds very little UPF1. Thus, there is the risk that one normalizes against essentially unspecific background signal. The authors should use instead a construct for normalization that is known to associate abundantly with UPF1.

4) Figure 8: The data for UPF3b is not convincing. In panel A, the increase in RNA abundance between NT and siUPF1 is about the same as the difference between siPTBP1 and siPTBP1/siUPF1, suggesting that the UPF1-induced increase is independent of PTBP1. Furthermore, panel B should be completed with a determining the half-lives of these two constructs under PTB and UPF1 knockdown.

---

## [Author Response]

*Essential revisions: 1) The effects are relatively weak. One way to strengthen the conclusions will be to insert PTBP1 binding sites downstream of nonsense codons in well characterized NMD reporters (e.g., β-globin or TCR-β), that are efficiently degraded by NMD or to tether PTBP1 downstream of a stop codon.* In the revised manuscript, we further bolster the evidence that PTBP1 is a potent inhibitor of NMD in two ways. First, we show that the RSE is able to abrogate NMD stimulated by the exon junction complex, using an established NMD reporter undergoing efficient decay due to a spliced intron downstream of the termination codon (Figure 1—figure supplement 2; GAPDH-AdML reporter from Singh et al. 2008). We employed this reporter after initial attempts to incorporate PTBP1 binding sites downstream of the classic “β-globin 39” premature termination codon led to aberrant splicing. As both the β-globin 39 and GAPDH-AdML reporters undergo NMD stimulated by a single downstream intron, these reporters are functionally similar. Second, we demonstrate that PTBP1 recruited via the bacteriophage MS2 coat protein stabilizes NMD target mRNAs (Figure 5—figure supplement 1). Further, it is important to note that we show that the RSE/PTBP1 can completely protect two well-studied, physiologically important NMD targets (SMG5 and the Rous sarcoma virus unspliced mRNAs) from decay.

*2) The evidence for the competition for mRNA binding model relies on the RNA-IP data. However, no X-linking was involved in the protocol and the authors cannot rule out about post-lysis rearrangements possibly confounding the results (as described by Mili and Steitz). The authors should perform control experiments to test the extent of such rearrangements in their assays.* To address the possibility that the observed ability of PTBP1 to inhibit UPF1 binding to mRNAs was due to RNA-protein interactions formed in cell extracts, we performed experiments in which extracts containing tagged UPF1 were mixed with extracts produced from cells expressing exogenous reporter mRNAs (Figure 3—figure supplement 1). Whereas affinity purification of tagged UPF1 efficiently recovered co-expressed mRNAs, we observed no co-purification of UPF1 with mRNAs expressed in distinct cells. These data suggest that the interactions assayed by immunoprecipitation reflect genuine cellular interactions and are not the result of post-lysis re-assortment.

*3) To use the GFP-bGH construct for normalization seems inadequate, since the authors mention that this binds very little UPF1. Thus, there is the risk that one normalizes against essentially unspecific background signal. The authors should use instead a construct for normalization that is known to associate abundantly with UPF1.*

For the post-lysis mixing experiments described in 2), we also used mRNAs containing a longer, NMD-sensitive artificial GAPDH 3’UTR as the basis for calculating relative recovery of mRNAs in the presence and absence of PTBP1 binding sites. This approach resulted in quantification of relative recovery of RSE variant mRNAs relative to each other and to the unmodified GAPDH 3’UTR indistinguishable from that observed in experiments using the shorter bGH-derived 3’UTR. In addition, we have previously demonstrated that using the GFP-bGH mRNAs as recovery controls gives results that are highly consistent with the orthogonal approach of tagged mRNA affinity purification (see Hogg and Goff, 2010).

4) Figure 8: The data for UPF3b is not convincing. In panel A, the increase in RNA abundance between NT and siUPF1 is about the same as the difference between siPTBP1 and siPTBP1/siUPF1, suggesting that the UPF1-induced increase is independent of PTBP1. Furthermore, panel B should be completed with a determining the half-lives of these two constructs under PTB and UPF1 knockdown.

Due to the relatively small effects of deleting putative PTBP1 binding sites from the UPF3B 3’UTR and the technical difficulty of mRNA decay assays incorporating RNAi-mediated protein depletion, we were unable to use this approach to conclusively demonstrate that the observed differences in mRNA stability were due to protection from NMD. As this point is not central to the manuscript, we have removed the previous Figure 8 pending further study.